# Resurgence of Influenza Circulation in the Russian Federation during the Delta and Omicron COVID-19 Era

**DOI:** 10.3390/v14091909

**Published:** 2022-08-29

**Authors:** Anna Sominina, Daria Danilenko, Andrey Komissarov, Ludmila Karpova, Maria Pisareva, Artem Fadeev, Nadezhda Konovalova, Mikhail Eropkin, Kirill Stolyarov, Anna Shtro, Elena Burtseva, Dmitry Lioznov

**Affiliations:** 1Smorodintsev Research Institute of Influenza, 197376 Saint Petersburg, Russia; 2National Research Center for Epidemiology and Microbiology Named after N.F. Gamaleya, 123098 Moscow, Russia; 3Department of Infectious Diseases and Epidemiology, First Pavlov State Medical University, 197022 Saint Petersburg, Russia

**Keywords:** influenza, respiratory viruses, SARS-CoV-2, surveillance, incidence, hospitalization, antigenic properties, NGS sequencing, susceptibility to antivirals

## Abstract

Influenza circulation was substantially reduced after March 2020 in the European region and globally due to the wide introduction of non-pharmaceutical interventions (NPIs) against COVID-19. The virus, however, has been actively circulating in natural reservoirs. In summer 2021, NPIs were loosened in Russia, and influenza activity resumed shortly thereafter. Here, we summarize the epidemiological and virological data on the influenza epidemic in Russia in 2021–2022 obtained by the two National Influenza Centers. We demonstrate that the commonly used baseline for acute respiratory infection (ARI) is no longer sufficiently sensitive and BL for ILI incidence was more specific for early recognition of the epidemic. We also present the results of PCR detection of influenza, SARS-CoV-2 and other respiratory viruses as well as antigenic and genetic analysis of influenza viruses. Influenza A(H3N2) prevailed this season with influenza B being detected at low levels at the end of the epidemic. The majority of A(H3N2) viruses were antigenically and genetically homogenous and belonged to the clade 3C.2a1b.2a.2 of the vaccine strain A/Darwin/9/2021 for the season 2022–2023. All influenza B viruses belonged to the Victoria lineage and were similar to the influenza B/Austria/1359417/2021 virus. No influenza A(H1N1)pdm09 and influenza B/Yamagata lineage was isolated last season.

## 1. Introduction

The growth of influenza surveillance and research in the past 10 years has generated substantial new data on the epidemiology and risk from influenza around the world; however, data on influenza disease burden remain scarce in middle- and low-income regions [1]. Influenza has been the main cause of annual epidemics of respiratory illness globally with a clear seasonal pattern in the Northern and Southern Hemispheres and year-round activity in the tropics [2]. The ever-changing nature of the virus necessitated its continuous surveillance, which was introduced globally in 1947 and is currently operated by the WHO Global influenza Surveillance and Response System (GISRS) [3]. One of the key components of the GISRS is the global network of the National Influenza Centers (NICs) that operate in accordance with Terms of Reference for the NICs [4] as well as the Pandemic Influenza Preparedness Framework [5].

The Russian Federation has a long-standing established comprehensive non-sentinel surveillance system based on the reporting of all positive influenza cases. For a more detailed description of influenza surveillance systems in Russia, see Reference [6]. In 2020, the system was adapted to include the SARS-CoV-2 data to monitor the progression and evolution of the COVID-19 pandemic and to provide critical information for public health decision-making. The study of genomic epidemiology of the early stages of outbreak in Russia provided understanding of the emergence of SARS-CoV-2 in Russia in March and April 2020 [7]. The COVID-19 emergency had a profound negative effect on influenza surveillance in many countries and WHO regions. Since the middle of 2020, the ECDC/WHO Regional Office for Europe outlined the need for the integration of SARS-CoV-2 into existing surveillance systems [8], and the WHO published global guidance on integrated sentinel surveillance in March 2022 [9]. The necessity of year-round integrated surveillance based on sentinel systems was indicated [10].

The emergence of this new pathogen with an immense pandemic potential has led to an unprecedented decrease in influenza activity globally since March 2020 [11,12]. In Russia, influenza activity in 2020–2021 was very low, as was the case elsewhere. Although COVID-19 activity in 2020 and early 2021 was lower [13] than that during the following Delta (since April 2021) and Omicron (since January 2022) waves, the NPIs in 2020 and the first half of 2021 were more stringent due to the low exposure to the virus and COVID-19 vaccination rates at the population level; COVID-19 vaccination was widely introduced only in January 2021. In summer 2021, COVID-19 activity in Russia stabilized, and many of the NPIs were loosened. Simultaneously, the first cases of laboratory-confirmed influenza were registered, and influenza activity progressed in Autumn and the following winter.

Here we present the results of the monitoring for influenza in Russia for season 2021–2022 as well as COVID-19 and other respiratory viruses. In order to provide clear evidence of SARS-CoV-2 impact on the influenza activity in the country in recent years, we also provide a brief comparison with the previous epidemic periods from 2015–2016 to 2021–2022. 

## 2. Materials and Methods

### 2.1. COVID-19 Containment Measures

The fast-evolving nature of the COVID-19 pandemic and the significant unknowns coming with a new virus and new disease have led to unprecedented challenges for healthcare systems as well as to dramatic socioeconomic impacts around the world. Many countries have introduced various restrictions in different waves of COVID-19 activity. In Russia, epidemiological monitoring for SARS-CoV-2 was introduced shortly after the discovery of a new pathogen in China. An influenza surveillance system was adapted for COVID-19 monitoring and sharing information with the WHO. Expanded real-time PCR testing capacities were introduced shortly after the declaration of pandemic in all certified laboratories, hospitals and primary and community care testing facilities, accessible for all risk groups as well as symptomatic patients and close contacts. The NPIs in 2020 included compulsory use of medical and personal protective equipment, restriction of mass gatherings, isolation of patients with COVID-19, PCR-testing of everyone returning from abroad, mandatory temperature monitoring, social distancing, school and university closure and a switch to online education and remote work for every possible position. The NPIs were subsequently loosened or toughened again depending on the intensity of COVID-19 circulation. In the described period in summer 2021, NPIs included compulsory mask use in all public transport and public places, mandatory vaccination of 80% of staff working at governmental organizations, QR codes of vaccination for entry to any public place, part-closure of educational facilities and non-essential shops as well as social distancing. By October-November 2021, these were complemented by a ban on mass gatherings (more than 500–1500 persons) and the closure of food courts and large malls. By March 2022, the mass gatherings could be resumed along with opening of shops and malls and the partial freedom from use of QR codes for entry. From spring 2022, the NPIs were gradually loosened until summer 2022, when all COVID-19 restrictions were lifted including wearing of protective masks and social distancing. 

### 2.2. Epidemiological Surveillance

The non-sentinel influenza surveillance system operates year-round in 60 major cities of Russia and collects weekly aggregated data on incidence, hospitalizations and deaths, along with laboratory confirmation for influenza, SARS-CoV-2, RSV and other respiratory viruses. The data were collected for the age groups 0–2, 3–6, 7–14, 15–64 and 65+. For the purposes of monitoring of influenza activity and determination of the start of the influenza epidemic and its end the baselines for ARI incidence, ARI hospitalization as well as clinically diagnosed influenza incidence and hospitalization are calculated on an annual basis. Such annual refinement is necessary for more accurate assessment of influenza activity. In 2021–2022 the baseline (BL) for incidence and BL for hospitalization with influenza were updated, taking into account changes of these indicators for the previous seven seasons from 2015–2016 to 2021–2022 period. Pre-epidemic and post-epidemic BLs were calculated using the method of moving epidemics [14]. 

### 2.3. PCR Detection of Influenza and Other Respiratory Viruses

RNA was isolated from clinical samples using the AmpliSense^®^RIBO-prep kit (InterLabService, Moscow, Russia) or the RNeasy Mini kit (QIAGEN). PCR for influenza A and B viruses was performed using the AmpliSense^®^ Influenza virus A/B-FL kit (InterLabService, Russia), and subtype identification was performed using AmpliSense^®^ Influenza virus A-FL kit subtyping H1N1, H3N2 (InterLabService, Russia) or the CDC Human Influenza Virus Real-Time RT-PCR Diagnostic Panel (Influenza A/B Typing Kit, Catalog #FluIVD03-1), CDC Human Influenza Virus Real-Time RT-PCR Diagnostic Panel (Influenza A Subtyping Kit, Catalog #K132508) or CDC Human Influenza Virus Real-Time RT-PCR Diagnostic Panel (Influenza B Lineage Genotyping Kit, catalogue #190302) with SuperScript III. The Multiplex AmpliSense^®^ kit for detection of influenza and SARS-CoV-2 was implemented as well. Detection of other respiratory viruses was performed using the multiplex AmpliSense^®^ ARVI-screen-FL kit (InterLabService, Russia). All real-time PCR tests were carried out using a Rotor-Gene 6000 (Corbett Research, Australia) or a CFX96 Touch™ Real-Time PCR Detection System (Bio-Rad, Hercules, CA, USA).

### 2.4. Influenza Virus Isolation, Identification, and Antigenic Analysis

Influenza virus isolation, identification, and antigenic analysis were performed according to the WHO Manual [15] and appropriate guidelines approved in Russia [16]. Influenza viruses were isolated in MDCK or MDCK-SIAT1 cells (MDCK cells transfected with the gene of 2,6-sialyltransferase), which were purchased by the Smorodintsev Research Institute of Influenza from the Institute of Virology, Marburg, in 2020 [17]. Madin-Darby canine kidney cells were obtained from the WHO CC World Influenza Centre, London. Influenza A and B viruses were isolated from rRT-PCR positive samples. Briefly, cells were seeded on Nunc cell culture tubes (Nunc, Thermo Fisher Scientific, Waltham, MA, USA) 1 day before inoculation to form a 90–95% confluent monolayer. Cells were washed twice with MEM media containing 2 μg of TPCK–trypsin and penicillin–streptomycin (10,000 units and 10 mg/mL, respectively), and 200 μL of virus-containing media was inoculated into each tube. Tubes were kept at 36 °C for 40 min to allow virus absorption. Then, 1.8 mL of virus growth media was added (MEM containing 2 μg of TPCK–trypsin and penicillin–streptomycin, bovine albumin fraction V (2.6 mL per 100 mL, Sigma-Aldrich, St. Louis, MO, USA), HEPES buffer (1.6 mL, Sigma-Aldrich)). Tubes were kept for 3–6 days at 36 °C and monitored daily for progression of CPE. Once CPE was detected, the tubes were frozen at −80 °C and thawed, and the hemagglutination assay with human (group O) red blood cell suspension (0.75%) was performed to determine viral titers. 

The viruses isolated during the epidemic period 2021–2022 were characterized antigenically in a hemagglutination inhibition test (HI) or in microneutralization assay (MN) with the strain-specific postinfectious ferret antisera kindly provided by Dr. John McCauley (WHO CC at the Francis Crick Institute in London) or strain-specific hyperimmune rat antisera to the reference influenza viruses.

*The MUNANA assay* for assessment of influenza viruses’ sensitivity to neuraminidase inhibitors was performed as according to the WHO Manual [15].

### 2.5. NGS Sequencing

Libraries for Illumina sequencing were prepared using the Nextera XT library preparation kit (Illumina, San Diego, CA, USA) and then sequenced on a MiSeq instrument (Illumina, USA) with a MiSeq Sequence kit v3. FastQC software was used for sequence data quality assessment. Trimmomatic was used for quality data trimming. Reads were mapped onto reference sequences using BWA. The consensus sequence was obtained using Samtools mpileup. Libraries for Oxford Nanopore sequencing were prepared using an SQK-LSK109 DNA Ligation Sequence kit (Oxford Nanopore, Oxford, UK). Sequencing was performed using a MinIon instrument (Oxford Nanopore, UK) with a R9.4.1 flowcell. Guppy software was used for basecalling and data quality trimming. Reads were mapped onto the reference sequence using Minimap2. 

### 2.6. Phylogenetic Analysis

Phylogenetic trees were built for the HA and NA genome segments. Reference human influenza virus sequences were downloaded from the EpiFlu database for analysis. Sequences for each genome segment were aligned using the Muscle application within MEGA7.0 software. Maximum-likelihood phylogenetic trees were built using MEGA7.0 with 1000 bootstraps using the Hasegawa–Kishino–Yano (HKY85) model with gamma rate categories for hemagglutinin sequences and Tamura (T92) model with gamma rate categories for neuraminidase sequences, respectively. 

### 2.7. Ethical Aspects of the Study

The study was performed in accordance with the Principles of Good Clinical Practice (GCP). The local ethics committee approved the study before initiation. Patient consent to study involvement was a research prerequisite.

## 3. Results

### 3.1. Non-Sentinel Epidemiological Surveillance Data

Monitoring the ILI incidence (excluding ARI and COVID-19) in the season 2020–2021 showed that after a period of “fluctuation” (from weeks 40 to 45 in 2021), when influenza activity remained near the baseline (BL), it began to increase steadily. The exceeding of the BL (0.06 per 10,000 of population) was registered on week 46 with a peak of influenza incidence at week 51 in 2021 (0.44 per 10,000), and afterwards the ILI rates began to decrease. By week 4 in 2022, the incidence of influenza fell below the epidemic BL and remained at a low level until the end of the season. 

In addition, an unusually early (from week 42 in 2021) increase in hospitalization rate for patients with ILI was observed, and the BL for hospitalization (0.04 per 10,000) was exceeded at week 47 in 2021. Hospitalization peaked (0.16 per 10,000 population) at weeks 50 to 51 of 2021 with a subsequent decrease and fell below the BL, as well as morbidity, in the 4th week of 2022. The intensity of the epidemic, assessed by comparison with the corresponding BLs of hospitalizations, appeared below the moderate intensity threshold (0.23 per 10,000), which indicated a relatively low severity of influenza cases in that season (Figure 1).

The comparative analysis of the last influenza epidemic with the previous seven epidemic periods revealed the limited possibility of determining the start and duration of influenza epidemics by analyzing the cumulative incidence of ILI and ARI due to the high impact of COVID-19 on the ARI incidence in the country (Figure 2).

This was a consequence of the fact that in the last two influenza seasons, mild and undiagnosed cases of COVID-19 were registered together with seasonal ARI, which led to an overestimation of the cumulative incidence.

In order to determine the timing of the influenza epidemic in the period of its simultaneous circulation with SARS-CoV-2, we evaluated the possibility of using the incidence and hospitalization data exclusively with clinically diagnosed influenza by comparison with corresponding BLs. It appeared that this approach was successful in determining the start and duration of influenza epidemics not only for the 2021–2022 season but also for the previous 7-year period. Intensity of the epidemic by incidence data was slightly above the BL, showing moderate level in the 2021–2022 season. This approach also confirmed the absence of an influenza epidemic in the 2020–2021 season. Taking into account only the ILI incidence, we confirmed the very high intensity of the epidemic in the 2015–2016 season, the low level of intensity of the epidemics in 2016–2017 and 2017–2018 and the moderate level of intensity of influenza epidemics in 2018–2019, 2019–2020 and 2021–2022 (Figure 3).

In 2021–2022, two waves of respiratory incidence were observed in Russia: The first one coincided with the influenza epidemic and the second one with the pronounced increase in the COVID-19 incidence caused by the Omicron variant. These waves of incidence replaced each other, emphasizing the widely discussed effect of viral interference. Thus, due to the active circulation of SARS-CoV-2, it is impossible to determine the boundaries of the influenza epidemic using the BL for the cumulative influenza and ARI incidence. Only use of the BL for ILI incidence made it possible to determine the main epidemiological parameters of the influenza epidemic (Figure 4).

### 3.2. PCR Monitoring of Influenza Activity

The etiology of ILI and ARI monitored in non-sentinel influenza surveillance based on weekly analysis of PCR detection of influenza and other respiratory viruses was performed at the two NICs and their 60 collaborating regional base laboratories. In total, 137,161 patients of different age groups were examined by PCR during the reporting period, and 8637 (6.3%) samples were positive for influenza. Among them, 7677 (88.9% of positive cases) were due to influenza A(H3N2) virus, six (0.1%) cases were associated with influenza A(H1N1)pdm09 and 275 (3.2%) cases were caused by influenza B viruses.

The first sporadic cases of influenza A(H3N2) were registered unusually early—in the summer—on weeks 33–36 of 2021. Since week 40 in 2021, the number of influenza-positive A(H3N2) cases began to increase and reached the 10% epidemic threshold unusually early—at week 47 in 2021, which was consistent with epidemiological data. The rate of influenza progressively increased, peaking (22.7%) at week 51 in 2021. The virus detection rate began to decline from week 2 in 2022 and dropped below the 10% epidemic threshold at week 3 in 2022 (8.0%). The influenza B detection rate was low throughout the whole period of the epidemic, with only a slight increase towards the end of the season.

According to laboratory-confirmed influenza (LCI) cases, the duration of the influenza epidemic in 2021–2022 in Russia was assessed as 8 weeks (from week 47 in 2021 to week 2 in 2022), although the sporadic circulation of influenza viruses was observed up to the end of the season. Influenza A (H3N2) viruses were the dominant agent during the whole epidemic. Influenza B viruses of Victoria lineage were detected sporadically with a more regular pattern towards the end of the season—from week 12 to week 20 in 2022 (Figure 5). The dominance of the influenza A(H3N2) virus was observed in all federal districts of the Russian Federation without exception.

The comparison of laboratory data with the epidemiological data showed a coincidence in timing of the peak of PCR influenza detection with the incidence and hospitalization peaks of patients with a clinical diagnosis of influenza. However, the duration of the influenza epidemic according to epidemiological criteria (incidence and hospitalizations above the BL) was somewhat longer, which could be the result of the registration of non-laboratory-confirmed, clinically mild COVID-19 cases, diagnosed as ILI.

### 3.3. Contribution of Other Respiratory Viruses

An additional analysis of the impact of seven non-influenza respiratory viruses (ORV) and SARS-CoV-2 for the ARI incidence at various stages of the epidemic process was performed. As a result of investigation of 129,054 patients with acute respiratory viral infections, some regularities in the circulation of pathogens were identified. An important role of rhinovirus and RS-virus infection in the incidence of acute respiratory viral infections in the pre-epidemic influenza period (from week 40 in 2021 to week 46 in 2021) was observed. During the epidemic period (from week 47 in 2021 to week 2 in 2022), the participation of seasonal respiratory viruses decreased markedly. Over the entire period, SARS-CoV-2, as expected, made the greatest contribution to the respiratory morbidity. A distinct increase in SARS-CoV-2 Omicron activity, registered since the first week of 2022, was accompanied by a steady decrease in the frequency of influenza virus detection. At week 6 in 2022, when the incidence of COVID-19 reached its peak (67% of investigated patients), influenza activity fell below 1% of all investigated cases.

Starting from week 12 in 2022, SARS-CoV-2 activity began to decrease with a simultaneous increase in cumulative activity of other respiratory viruses (from 4.5% at the peak of SARS-CoV-2 to 13.2–15.4% on last weeks of the season) when the percent of other laboratory-confirmed respiratory viruses exceeded that for SARS-CoV-2 (Figure 6).

An analysis of the age-specific peculiarities of the etiology of acute respiratory morbidity under routine surveillance showed that influenza viruses affected all age groups of the population, while other respiratory viruses (ORV), especially RSV, were predominantly registered in young children under 6 years of age, where the frequency of detection of these infections was 4–9 times higher than in the age group ≥ 15 years. The most affected by all ORV (parainfluenza viruses, adenoviruses, RSV, bocaviruses, metapneumoviruses, seasonal coronaviruses and rhinoviruses) were newborns and infants ≤ 2 years. With age, the frequency of detection of these viruses progressively decreased. These data serve as an explanation for such a high incidence of ARI in young children, recorded in epidemiological observations. The overall frequency of detection of influenza in the last season was 6.3%, while for non-influenza viruses (excluding SARS-CoV-2) the frequency of detection was 9.6%. SARS-CoV-2 remained the dominating agent causing 34.3% of overall cases of respiratory viral diseases with the greatest impact in the adult population (41.7% of all positive ILI/ARI cases). Schoolchildren aged 7–14 years were affected by SARS-CoV-2 less often (by 1.7 times) and children of younger age groups 0–2 and 3–6 years by 2.8–3.3 times less often compared to adults (Table 1).

Influenza virus isolation and characterization were performed in two NICs and 25 RBLs. A total of 1734 PCR influenza-positive clinical samples were investigated in the current season, and 499 (28.8%) influenza viruses were isolated, including 471 influenza A(H3N2) viruses and 28 influenza B viruses. The dominant causative agent of the epidemic was influenza A(H3N2) virus (94.4% of the total number of isolates). The number of isolated viruses decreased in this season, which was largely due to the reorientation of laboratories to the surveillance for the circulation of the SARS-CoV-2 pandemic virus.

The first strains of the influenza A (H3N2) virus were isolated at weeks 43–45, 2021, from samples collected on these weeks or one week before. The number began to increase from week 46, 2021, with a peak of isolation at week 52, 2021, when the percentage of virus isolation reached 64.4% of the number of samples examined, with the absolute dominance of the A(H3N2) virus. The first strains of the influenza B virus were isolated only on week 4 in 2022, and the number of isolated viruses of this type for the entire season was low (5.6% of the total number of isolates).

The results of the isolation of influenza viruses are shown in Figure 7. The dominance of the influenza A (H3N2) virus throughout the entire epidemic cycle is clearly visible, with a slight increase in the proportion of influenza B viruses in the second half of the epidemic. Influenza viruses continued to be isolated until the end of the study period; however, no influenza A(H1N1)pdm09 virus was isolated due to their low epidemic activity.

### 3.4. Susceptibility of Influenza Viruses to Antivirals

288 isolated influenza viruses were analyzed for susceptibility to neuraminidase inhibitors (oseltamivir and zanamivir) using the MUNANA fluorescence assay (Table 2). Of these, 276 strains belonged to subtype A(H3N2), and 12 strains were influenza B of the Victoria lineage. Influenza A(H3N2) viruses were obtained from nine territorially remote cities of the Russian Federation and influenza B from two cities. All influenza A(H3N2) viruses were susceptible to both neuraminidase inhibitors. One influenza B virus (B/St. Petersburg/NIIG-03/2022) had reduced susceptibility to oseltamivir.

### 3.5. Antigenic Analysis of Isolates

The majority of 193 influenza A(H3N2) viruses were antigenically similar to the vaccine strain A/Darwin/9/2021, recommended for the influenza vaccine composition for the season 2022–2023, and showed deviation from the strain A/Cambodia/e0826360/2020 recommended for the season 2021–2022. All 12 tested influenza B viruses circulating in Russia belonged to the Victoria lineage and were antigenically similar to the reference influenza B/Austria/1359417/2021 virus, also recommended by the WHO for the composition of vaccines for the upcoming season for the Northern Hemisphere.

### 3.6. Genetic Analysis of Influenza Viruses

Most influenza A(H3N2) viruses were quite genetically homogeneous and belonged to the clade 3C.2a1b.2a.2 with characteristic amino acid substitutions in HA, such as K83E, Y94N, T131K (-CHO), F193S, Y195F and I522M. At the same time, Russian viruses differed from vaccine strain A/Cambodia/e0826360/2020 in terms of substitutions of amino acid residues in HA: Y159N, T160I, L164Q, G186D and D190N. Despite the homogeneity, several genetic groups with characteristic substitutions were distinguished within the clade. More than 80% of Russian viruses belonged to the subgroup of influenza A/Darwin/9/2021(H3N2) virus recommended by the WHO for the influenza vaccines for the 2022–2023 epidemic season in the Northern Hemisphere with a characteristic substitution H156S in the HA1 subunit.

Phylogenetic analysis showed that influenza B viruses belonged to the genetic group 1A-Δ3, B/Washington/02/2019-like viruses carrying residues K162, N163 and N164 in HA1. Influenza B viruses clustered into a genetic subgroup V1A.3a.1, which includes influenza B/Austria/1359417/2021 virus recommended by the WHO for trivalent and quadrivalent influenza vaccines for the 2022–2023 epidemic season in the Northern Hemisphere. Phylogenetic analysis of HA of influenza B viruses sequenced in Russia revealed that they formed one genetic subgroup with a 6 amino acid substitution, including two positions in the antigenic sites BA (N150K) and BB1 (N197D), the latter leading to the loss of a potential glycosylation site. Viruses of this subgroup were isolated in three different regions of the Russian Federation (St. Petersburg, Khabarovsk, Moscow).

## 4. Discussion

The COVID-19 pandemic has caused lasting disruption in the global influenza surveillance, particularly to sentinel systems, and considerable time and investment will be required in many countries before expanded, sustainable systems can be re-established. The ongoing COVID-19 pandemic remains unpredictable, with many countries planning to continue the collection, reporting, and use of non-sentinel COVID-19 data in 2022/23 to inform policy decisions.

Global influenza surveillance, implemented by NICs within the WHO system, is of great importance for strengthening public health around the world. Due to the high ability of transmission, influenza viruses, and now also the SARS-CoV-2 virus, do not know borders and primarily affect countries with extensive international traffic and human migration [18]. The high variability of these pathogens allows them to overcome the established immunity and cause annual epidemics [19]. Strains with altered antigenic and genetic properties regularly appear in one country or another, and the place of their appearance is not predictable, as is the situation with the dominance in the coming season of one of the many antigenic variants among viruses circulating in the Northern and Southern Hemispheres.

Here we described the resurgence of influenza circulation in Russia after the non-epidemic season of 2020–2021 when the very low influenza activity was registered in the majority of countries around the world. Such low activity was primarily explained by the introduction of NPIs, and many of the countries have described the reduction in influenza activity in 2020–2021 compared to previous seasons [20,21,22]. Interestingly, influenza activity has resumed earlier at some regions and countries where the COVID-19 activity was low for a prolonged period of 2020–2021 (before the Omicron wave). For example, China has had the most stringent NPIs and zero COVID-19 policy since the emergence of the virus in late December 2019. In 2020, they had a pronounced decrease in influenza activity as observed elsewhere. However, in the beginning of 2021 the influenza B activity started to rise as in Northern as well as in Southern China and was causing a full-scale influenza epidemic in autumn 2021 [23], despite the NPIs being in place for the whole period of monitoring.

Influenza activity in the WHO European region in 2021–2022 was registered in two waves. Influenza virus activity began to increase from week 48 in 2021 and was driven predominantly by influenza A(H3N2) viruses. This was very similar to what we saw in Russia. The peak of influenza activity in Europe was recorded later—at week 52, 2021 (19% of positive samples), after which the number of samples positive for influenza began to decrease. Meanwhile, a second rise of influenza activity was registered from week 9 in 2022, which reached its maximum at week 12 in 2022 (30% of positive samples from the number of those examined), which was atypically late for an influenza epidemic in Europe [24]. The two waves of influenza activity in Europe were split by the SARS-CoV-2 Omicron wave of different subgroups, while in Russia the Omicron wave came later and did not split the influenza epidemic. The two-wave incidence of influenza was also registered in the US [25].

In the majority of countries of the Northern Hemisphere, the 2021–2022 influenza season was dominated by the A(H3N2) viruses, as we saw in Russia [26]; however, the COVID-19 pandemic might have a profound impact on the sensitivity of influenza surveillance on a global scale. Meanwhile, some countries exhibited the opposite. The influenza epidemic of 2021–2022 in China was caused by influenza B of Victoria lineage which was the principal etiological agent, while in South Africa the season of 2022 was primarily caused by the A(H1N1)pdm09 virus [27]. This diversity shows that all known influenza subtypes are circulating at different levels around the globe, and constant monitoring is needed in order to be able to detect the emergence of any new antigenically distinct variants in a timely manner. This is especially important for vaccine strain selection.

The effectiveness of influenza vaccines depends on such a “hit on the target”. Selected strains, in addition to the above, must be sufficiently immunogenic and retain their antigenic properties during passage in chicken embryos at the stage of obtaining high-yielding reassortants and during vaccines reliant on egg production. Unfortunately, in some cases, viruses undergo host-dependent mutations, which distort the spectrum of antibodies induced by them, possibly reducing the protective properties of vaccines [28,29,30,31].

Nevertheless, vaccination remains the only method of reliable influenza prevention [32], since the widespread use of antiviral drugs for this purpose leads to the emergence of resistant variants of the virus, as happened with rimantadine. Therefore, it is important that further expansion of work on the isolation of influenza viruses and their antigenic and genetic characterization is carried out based on the WHO Collaborating Centers for Influenza and National Influenza Centers in different parts of the world to provide a wide panel of viruses for selection of the strain suitable for candidate vaccine strains.

The COVID-19 pandemic has emphasized the importance of well-established routine and sentinel surveillance systems for early recognition of a newly emerged pathogen on the country and global level, as well as the necessity of updating pandemic preparedness plans as the core activities in responding to new threats in a timely manner.

A comparison of influenza, ARI and COVID-19 incidence in the season 2021–2022 confirmed the previously obtained data on the interference between influenza viruses and SARS-CoV-2 during their circulation in the human population. Our analysis confirmed that the circulation of influenza viruses and RSV was suppressed, that influenza epidemics in most countries, including Russia, were not registered in the season 2020–2021 and that for the first time in recent decades, no influenza epidemic was registered in Russia [33], which was also observed in other countries of the world.

The low levels of influenza circulation at the global scale affected the level of population immunity to influenza viruses, which dropped sharply, and in this context as well as with the weakening of NPIs against COVID-19, early in the 2021–2022 season, influenza viruses returned to circulation. However, the intensity of the last epidemic in Russia was low. Moreover, after the New Year, unlike the previous 5 years, it sharply declined against the backdrop of a rapid increase in SARS-CoV-2 activity, and only sporadic influenza cases were still being registered in the country.

Extended monitoring of the etiology of respiratory diseases with the additional inclusion of pathogens of non-influenza ARI confirmed the pattern revealed in the 2020–2021 season, which manifests itself in a decrease in the circulation intensity of not only influenza viruses but also other pathogens (RSV, rhinovirus and adenovirus) during the period of increased activity of SARS-CoV-2. This interesting natural phenomenon still needs additional research to elucidate the mechanisms that regulate the range of circulation of various pathogens (interferon, the complement system, antiviral immunity or other factors).

Currently, epidemiological and virological components act together in order to develop approaches in the investigation of the biological properties of newly emerging influenza and other respiratory viruses circulating in the human population and to establish correlations between peculiarities of virus genome structure, pathogenicity and transmission capacity of these viruses in the human population according to the WHO Global Influenza Strategy 2019–2030 [34].

## 5. Limitations

The analysis presented in the paper contains certain limitations, the most important of which are the limited geographic representation and use of routine surveillance data only. In particular, only 60 regions of the country were covered by the influenza surveillance system established by the NICs out of 86 in the country. Limited by the incomplete geographical representativeness, we may have missed some circulation of influenza A(H1N1)pdm09 or influenza B, as well as minor epidemiological trends in circulation of non-influenza respiratory pathogens. Moreover, as routine surveillance has only aggregated data and does not separate severe patients from the general hospitalization statistics, it was impossible to assess the contribution of influenza of different subtypes, SARS-CoV-2 and other respiratory viruses to the severity of disease in the described period.

## Figures and Tables

**Figure 1 viruses-14-01909-f001:**
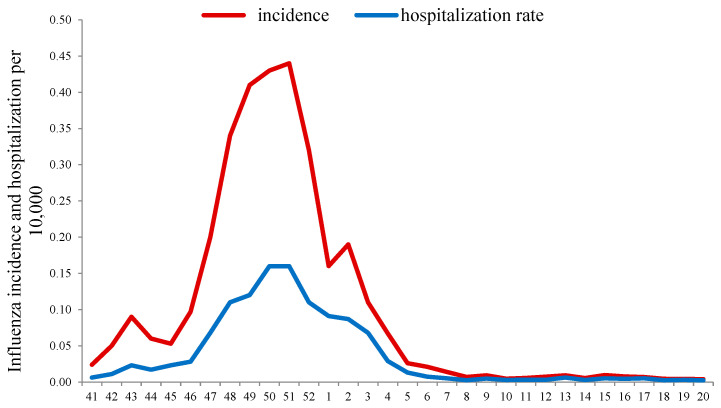
Dynamics of influenza incidence and hospitalization according to clinical diagnosis in the 2021–2022 influenza season in Russia.

**Figure 2 viruses-14-01909-f002:**
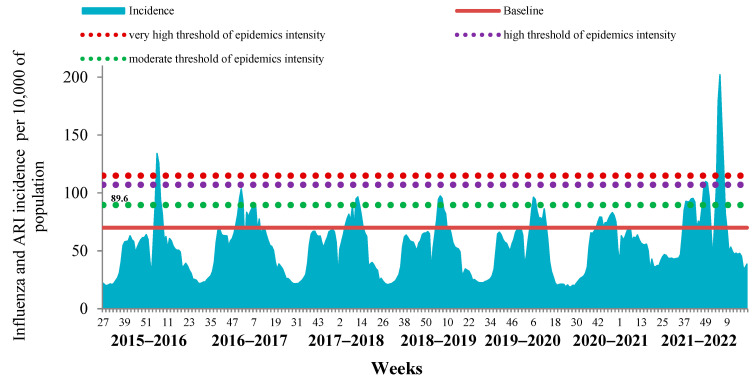
Dynamics of influenza and ARI cumulative incidence in the Russian Federation over the 7-year period.

**Figure 3 viruses-14-01909-f003:**
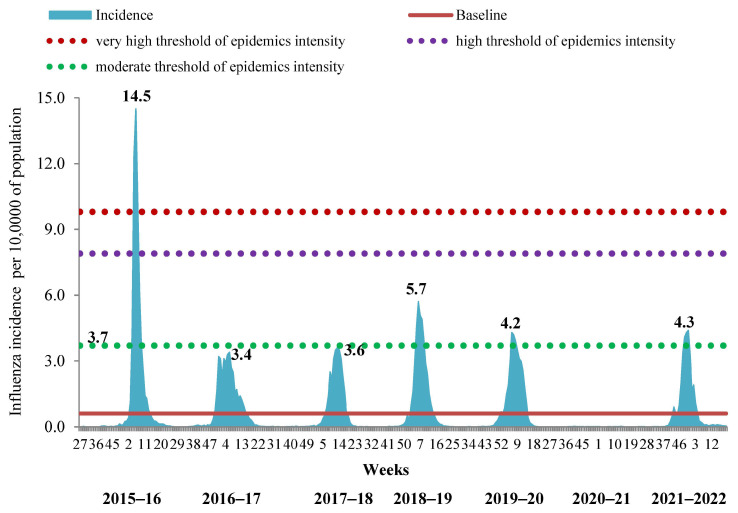
Assessment of the duration and intensity of influenza epidemics according to the incidence of clinically diagnosed influenza for the period from 2015 to 2022.

**Figure 4 viruses-14-01909-f004:**
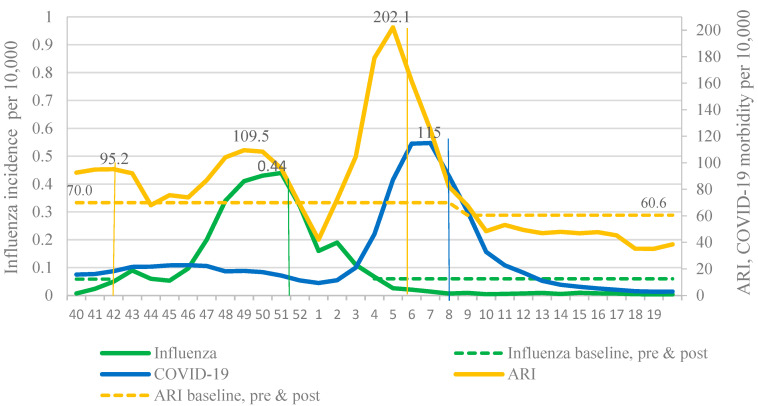
Comparison of the dynamics of the incidence of influenza, ARI and COVID-19 in the cities of the Russia under observation, 2021–2022 season.

**Figure 5 viruses-14-01909-f005:**
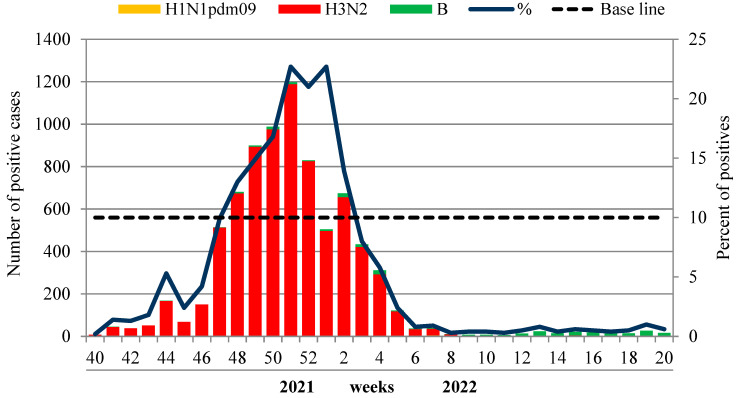
PCR-monitoring of influenza A(H3N2) and B virus circulation, season 2021–2022.

**Figure 6 viruses-14-01909-f006:**
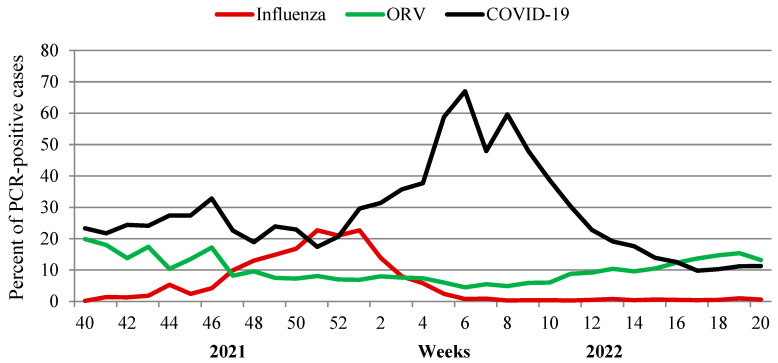
Detection of influenza, SARS-CoV-2 and other respiratory viruses in the Russian Federation according to results of non-sentinel surveillance (season 2021–2022).

**Figure 7 viruses-14-01909-f007:**
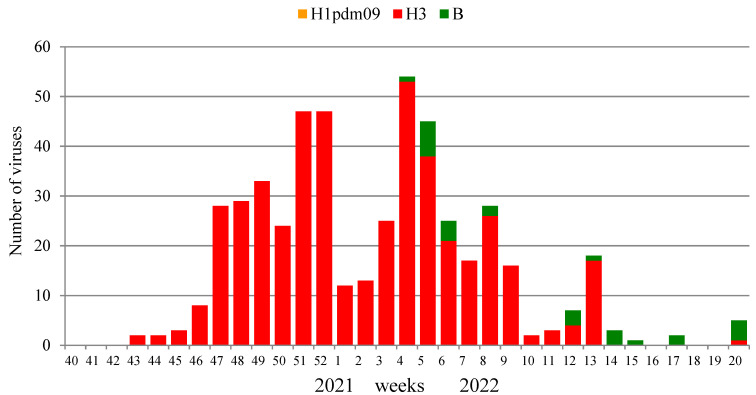
Monitoring of influenza virus isolation in two NICs and collaborating RBLs, season 2021–2022.

**Table 1 viruses-14-01909-t001:** Age-specific peculiarities of the etiology of respiratory viral infections according to PCR data under non-sentinel surveillance, season 2021–2022.

Age (Years)	Percent of Virus PCR Detection in Clinical Specimens													
	Influenza					Other Respiratory Viruses								SARS-CoV-2
	A	A (H1N1)pdm09	A(H3N2)	B	Total	PIV	AdV	RSV	Boca	MPV	s-CoV	Rhino	Total	
0–2	0.1	0.0	4.3	0.1	4.5	1.3	2.7	7.3	1.4	0.4	1.3	5.9	20.4	15.1
3–6	0.3	0.0	6.5	0.1	6.9	1.0	1.6	3.2	0.5	0.4	0.8	4.7	12.3	12.5
7–14	0.4	0.0	9.8	0.1	10.3	0.8	0.7	1.4	0.2	0.2	0.7	3.6	7.5	23.7
15 and more	0.7	0.0	4.5	0.3	5.5	1.0	0.5	0.8	0.2	0.2	0.8	2.5	5.9	41.7
Total	0.5	0.0	5.6	0.2	6.3	1.0	1.1	2.3	0.45	0.2	0.9	3.6	9.6	34.3

Note: PIV—parainfluenza virus, AdV—adenovirus, RSV—respiratory syncytial virus, Boca—bocavirus, MPV—metapneumovirus, s-CoV—seasonal coronavirus, Rhino—rhinovirus.

**Table 2 viruses-14-01909-t002:** Susceptibility of influenza viruses circulating in the Russian Federation to antivirals.

№	City	Number of Strains	Subtype	Value Range (IC_50_ µM)	
				Oseltamivir	Zanamivir
1	Astrakhan	6	A(H3N2)	0.5–1.2	0.5–2.7
2	Belgorod	1	A(H3N2)	0.8–0.8	0.7–0.7
3	Kaliningrad	6	A(H3N2)	0.8–1.4	0.6–1.3
4	Khabarovsk	16	A(H3N2)	0.3–1.6	0.2–2.4
5	Krasnoyarsk	24	A(H3N2)	0.6–2.6	0.7–1.9
6	Novosibirsk	36	A(H3N2)	0.6–3.5	0.4–2.6
7	Saint Petersburg	82	A(H3N2)	0.4–2.7	0.5–2.0
8	Samara	7	A(H3N2)	0.7–1.3	0.8–1.1
9	Stavropol	8	A(H3N2)	0.7–1.1	0.8–1
10	Saint-Petersburg	1	B/Victoria	213.0	2.0
11	Samara	1	B/Victoria	81.2	1.4
12	Moscow	28	A(H3N2)	0.2–3.4	0.5–3.5
13	V. Novgorod	11	A(H3N2)	0.2–0.6	0.4–1.4
14	Vladimir	5	A(H3N2)	0.3–0.6	0.6–1.2
15	Yaroslavl	24	A(H3N2)	0.3–1.0	0.5–1.2
16	Orenburg	8	A(H3N2)	0.1–0.7	0.3–2.5
17	Vladivostok	14	A(H3N2)	0.3–0.5	0.5–1.7
18	Moscow	10	B/Victoria	28.5–34.1	1.3–6.7

Note: Positions 1–11—data from the St. Petersburg NIC, positions 12–18—data of the Moscow NIC.

## Data Availability

All data on influenza epidemiological and virological surveillance in Russia in the season 2021–2022 and all the previous seasons described in the article are freely available as the weekly influenza bulletins at the website of Smorodintsev Research Institute of Influenza (https://www.influenza.spb.ru/en/influenza_surveillance_system_in_russia/epidemic_situation/, accessed on 31 July 2022).

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
