# Peer review of "Resurgence of Influenza Circulation in the Russian Federation during the Delta and Omicron COVID-19 Era"

_viruses, 2022, doi:10.3390/v14091909_

Round 1

Reviewer 1 Report

This manuscript is a nice description of influenza in Russia over the last year. The style and the choice of certain words needs improving. 

This is a nice report on the seasonal human influenza activity seen in large parts of the Russian Federation. My comments on the manuscript are essentially focussed on the use of English.

Line 17, replace ‘nature’ with ‘natural’.

Line 18, delete ‘the’ in front of NPIs.

Line 23, replace the word ‘in’ with the word ‘at’.

Lines 25 and 27, the numbers 21 should be 2021 describing the virus name – this is now the standard way of naming in influenza viruses. This might apply elsewhere also.

Line 35, replace the word ‘was’ with ‘has been’.

Line 40, pluralise the word ‘component’.

Line 42, capitalise the F of ‘framework’.

Line 43, replace the word ‘the’ with ‘a’.

Line 44, insert ‘a’ in front of the word ‘more’.

Line 56, replace ‘the new pathogen’ with ‘this new pathogen’ and replace ‘with the immense pandemic’ with ‘with an immense pandemic potential’.

Line 58, insert a comma after the word ‘low’.

Line 61, was it that there were low vaccination rates in the lead up to the 20 21/22 influenza season?

Line 62 delete the word ‘the’ in front of COVID-19.

Line 65, insert the word ‘the’ in front of ‘following winter’.

Line 74, insert a comma after the word ‘deaths’.

Line 76, insert the word ‘and’ in front of 65.

Line 77, reword to say ' the start of an influenza epidemic’.

Lines 79, re word to say ‘on an annual basis’.

Line 87, provide a reference for CDC primers and probes.

Line 111, the authors need to mention MDCK-SIAT1 cells in the preceding paragraph and provide a reference to these cells.

Line 130, it should be stated that the virus gene sequences were downloaded from the EpiFlu database, delete the word ‘the’ at the end of the line.

Line 132, it was not clear what point (4) was, and needs rewriting.

Line 144, the word ‘hesitated’ should be ‘remained’.

Lines 145 to 147, the meaning of this sentence was not clear and needs recomposing.

Line 149, is the word ‘medium’ the correct word?

Lines 152 to 153, re-word the section to read ‘at weeks 50 to 51 of 2021’.

Figure 1 legend, this should read ‘in the 2021 - 2022 influenza season in Russia’.

Lines 164 to 165, there should be no break in the paragraph.

Figures 2 and 3 graphs, the X axes should be marked better indicating the yearly transitions, and the title should be ‘thresholds of epidemic intensity’ not ‘thresholds of epidemics intensity’.

Line 177, insert the word ‘an’ in front of the word ‘influenza’.

Line 190, should read either ‘this wave’ or ‘these waves’.

Line 192, delete ‘the’ in front of SARS-CoV-2.

Line 232, needs to be recomposed to make better sense.

Line 238, ‘An’ needs to replace ‘the’ in front of important role.

Line 246, delete the word ‘the’ in front of ‘influenza’.

Line 248, delete the word ‘the’ in front of ‘SARS-CoV-2’.

Line 267, needs to be recomposed to make better sense.

Lines 288 to 293, it needs to be clear that it was the samples collected during the named weeks rather than being isolated in those weeks - this paragraph needs modification.

Line 306, there are two stops after the brackets around Table 2.

Line 323, insert the word ‘virus’ after the words ‘the reference’.

Line 334, the HA1 H156S substitution does not result in the loss or gain of a glycosylation site.

Line 342, the glycosylation motif at residues 197 to 199 of HA1 is known to be used, and so is a glycosylation site rather than a ‘potential glycosylation site’.

Line 348, I am not sure if this should be ‘established’ or ‘re-established’.

Line 355, I think the words ‘traffic flows’ does not convey the authors’ meaning.

Line 356, the word ‘highest’ probably should be replaced by the word ‘high’.

Line 377, it was not clear from the sentence which circulation was delayed.

Line 379, the word ‘the’ should be ‘a’.

Line 387, delete the word ‘it’.

Line 389, the word ‘the’ should be ‘a’.

Line 394, the word ‘the’ should be ‘any’.

Line 399, the authors should add ‘vaccines reliant on egg production’.

Line 401, I think the authors overstate the evidence that the protective properties of vaccines are reduced by mutation in the vaccine virus: there seems to me to be no conclusive proof of this and so this remains a possibility. I think it is best to describe this as ‘possibly reducing the protective properties of vaccines’.

Line 404, delete the word ‘it’.

Line 409, the words ‘becomes obvious’ are not needed.

Lines 437 to 438, insert the word ‘the’ in front of the words ‘investigation’, ‘biological’, and ‘human’, delete the word ‘the’ in front of ‘correlation’ and pluralize the word correlation.

Line 442 includes mention of the ‘agricultural sector’ and a ‘One-Health’ approach, but this manuscript really does not cover at all viruses at the animal-human interface. This either needs elaborating on or deleting.

Author Response

Dear Reviewer 1,

Thank you very much for your hard work reviewing our manuscript and for the precise comments provided. A special thank you for your patience to approve the language, we cordially express our gratitude for this.We have made all corrections to the English language in accordance with your recommendations. We also tried to modify and rephrase the sentences that were marked as unsuitable or hard to understand and tried to answer the questions posed.

  • All suggested corrections were inserted in the text. Below we provide point-by-point answers to the questions and rephrased sentences.

  • Reviewer's note, Line 61, was it that there were low vaccination rates in the lead up to the 20 21/22 influenza season?

Original version:

Although the COVID-19 activity in 2020 and early 2021 was lower than that during the following Delta (since April 2021) and Omicron (since January 2022) waves, the NPIs in 2020 and the first half of 2021 were more stringent due to the low exposure to the virus and vaccination rates at the population level [13]

Corrected version:

Although the COVID-19 activity in 2020 and early 2021 was lower [13] than that during the following Delta (since April 2021) and Omicron (since January 2022) waves, the NPIs in 2020 and the first half of 2021 were more stringent due to the low exposure to the virus and COVID-19 vaccination rates at the population level; COVID-19 vaccination was widely introduced only in January 2021.

  • Reviewer's note, Line 87, provide a reference for CDC primers and probes.

Original version:

Reverse transcription of RNA was performed using the Reverta- 86 L kit (InterLabService, Russia) or the OneStep RT-PCR Kit (QIAGEN) with CDC primers

and probes.

Corrected version:

PCR for influenza A and B viruses was performed using the AmpliSense® Influenza virus A/B-FL kit (InterLabService, Russia), subtype identification was performed using Am-pliSense® Influenza virus A -FL kit subtyping H1N1, H3N2(InterLabService, Russia) or the CDC Human Influenza Virus Real-Time RT-PCR Diagnostic Panel (Influenza A/B Typing Kit, Catalog # FluIVD03-1), CDC Human Influenza Virus Real-Time RT-PCR Di-agnostic Panel (Influenza A Subtyping Kit, Catalog # K132508), CDC Human Influenza Virus Real-Time RT-PCR Diagnostic Panel (Influenza B Lineage Genotyping Kit, catalogue #190302) with SuperScript III. 

Reviewer's note, Line 111, the authors need to mention MDCK-SIAT1 cells in the preceding paragraph and provide a reference to these cells.

Original version:

Influenza viruses isolated in MDCK or MDCK-SIAT during the epidemic period 2021- 2022 were characterized antigenically in hemagglutination inhibition test (HI) or in microneutralization assay (MN) with the strain- specific post infectious ferret antisera kindly provided by Dr. John McCauley (WHO CC at the Francis Crick Institute in London) or strain-specific hyperimmune rat antisera to the reference influenza viruses.

Corrected version:

Influenza virus isolation, identification, and antigenic analysis  were performed according to the WHO Manual [15] and appropriate guidelines approved in Russia [16]. Influenza viruses were isolated in MDCK or MDCK- SIAT1 cells (MDCK cells transfected with the gene of 2,6-sialyltransferase) which were purchased by Smorodintsev Research Institute of Influenza from the Institute of Virology, Marburg,  in 2020 [17]. Madin-Darby canine kidney cells (London line,was obtained from WHO CC World Influenza Centre, London). were used for isolation ofI influenza A and B viruses isolated  from rRT-PCR positive samples.

Additional link provided:

  1. Matrosovich, M.; Matrosovich, T.; Carr, J.; Roberts, N.A.; Klenk, H.-D. Overexpression of the α-2, 6-sialyltransferase in MDCK cells increases influenza virus sensitivity to neuraminidase inhibitors. J. Virol. 2003, 77, 8418–8425.

  • Reviewer's note, Line 132, it was not clear what point (4) was, and needs rewriting.

Original version:

The following criteria were used: (1) full genome segments; (2) virus type A and B; (3) A(H1N1)pdm09, A(H3N2) subtype; B Victoria and (4) reference of vaccine strain.

Corrected version:

Phylogenetic trees were built for the HA and NA genome segments. Reference human influenza virus sequences were downloaded from the EpiFlu database for analysis. Sequences for each genome segment were aligned using the Muscle application within MEGA7.0 software. Maximum-likelihood phylogenetic trees were built using MEGA7.0 with 1000 bootstraps using the Hasegawa-Kishino-Yano (HKY85) model with gamma rate categories for hemagglutinin sequences and Tamura (T92) model with gamma rate categories for neuraminidase sequences, respectively.

  • Reviewer's note, Lines 145 to 147, the meaning of this sentence was not clear and needs recomposing.

Original version:

The exceeding of BL (0.06 per 10 000 of population) was registered on the week 46 with a peak at week 51.2021 (0.44 per 10 000) and following decrease in incidence.

Corrected version:

The exceeding of BL (0.06 per 10 000 of population) was registered on the week 46 with a peak of influenza incidence at week 51.2021 (0.44 per 10 000) and afterwards the ILI rates began to decrease.

Reviewer's note, Line 149, is the word ‘medium’ the correct word?

Original version:

Intensity of epidemic by incidence data was slightly above the medium BL.

Corrected version:

Intensity of epidemic by incidence data was slightly above the BL showing moderate intensity.

  • Reviewer's note, Line 232, needs to be recomposed to make better sense.

Original version:

However, the epidemiological criteria determine the duration of the epidemic more extended; this could be the result of the contribution of non-identified COVID-19 cases, clinically similar to influenza.

Corrected version:

However, the duration of influenza epidemic by epidemiological criteria (incidence and hospitalizations above the BL) was somewhat longer, which could be the result of registration of non-laboratory confirmed, clinically mild COVID-19 cases, diagnosed as ILI.

  • Reviewer's note, Line 267, needs to be recomposed to make better sense.

Original version:

Last season, the frequency of registration of laboratory-confirmed influenza was generally lower than all ORV (6.3% and 9.6%, respectively), but SARS CoV-2 remained the leading virus detected as the cause of respiratory morbidity: it caused 34.3% of diseases among the entire population with the greatest impact on the adult population (41.7%).

 Corrected version:

The overall frequency of detection of influenza in the last season was 6.3% while for non-influenza viruses (excluding SARS-CoV-2) the frequency of detection was 9.6%.  SARS CoV-2 remained the dominating agent causing 34.3% of overall cases of respiratory viral diseases with the greatest impact in the adult population (41.7% of all positive ILI/ARI cases).

  • Reviewer's note, Lines 288 to 293, it needs to be clear that it was the samples collected during the named weeks rather than being isolated in those weeks - this paragraph needs modification.

Original version:

The first strains of the influenza A (H3N2) virus were isolated at 43-45 weeks. Their number began to increase from week 46, 2021, with a peak of isolation at week 52, 2021, when the percentage of virus isolation reached 64.4% of the number of samples examined, with the absolute dominance of the A(H3N2) virus.

Corrected version:

The first strains of the influenza A (H3N2) virus were isolated at weeks 43-45, 2021 from samples collected on these weeks or one week before. Their number began to increase from week 46, 2021, with a peak of isolation at week 52, 2021, when the percentage of virus isolation reached 64.4% of the number of samples examined, with the absolute dominance of the A(H3N2) virus.

  • Reviewer's note, Line 442 includes mention of the ‘agricultural sector’ and a ‘One-Health’ approach, but this manuscript really does not cover at all viruses at the animal-human interface. This either needs elaborating on or deleting.

Original version:

The future system must be built on inter-sectoral collaborations of the WHO NICs, agricultural sector and institutions and contribute to one health approach at animal-human interface.

Corrected version:  

This text has been removed.

The axes on graphs 2 and 3 were corrected to make them easier to read.

Yours sincerely,

Anna Sominina, Daria Danilenko and co-authors

Reviewer 2 Report

The article submitted for review is adequately structured and presented. Dealing with a topic of high interest to readers. As its fundamental core is the description of the epidemiology of the epidemiology of the influenza virus in a given period of time in which there was a COVID-19 pandemic, it would be interesting for the authors to describe in greater detail the isolation measures adopted in the country under study, since, as they themselves comment, there were containment measures in almost all countries, but not always the same and with the same rigorousness. I believe that this point is important for the reader to be able to make a comparative analysis with other articles in the literature describing the situation in other countries.

I believe that with this small modification, the article could be published in the journal. 

Author Response

Dear Reviewer 2,

Thank you very much for your hard work reviewing our manuscript and for the comments provided. Please see below our answer:

Reviewer 2 comment:

The article submitted for review is adequately structured and presented. Dealing with a topic of high interest to readers. As its fundamental core is the description of the epidemiology of the influenza virus in a given period of time in which there was a COVID-19 pandemic, it would be interesting for the authors to describe in greater detail the isolation measures adopted in the country under study, since, as they themselves comment, there were containment measures in almost all countries, but not always the same and with the same rigorousness. I believe that this point is important for the reader to be able to make a comparative analysis with other articles in the literature describing the situation in other countries.

Response:  We fully agree that description of NPIs in more detail was necessary. In this corrected version we inserted a separate paragraph describing in detail the isolation measures in Russia. The paragraph is inserted in “Materials and Methods”.

COVID-19 containment measures. The fast evolving nature of the COVID-19 pandemic and the significant unknowns coming with a new virus and new disease have led to unprecedented challenges for healthcare systems as well as to dramatic socio-economic impacts around the world. Many countries have introduced various restrictions in different waves of COVID-19 activity. In Russia, epidemiological monitoring for SARS-CoV-2 was introduced shortly after the discovery of a new pathogen in China. Influenza surveillance system was adapted for COVID-19 monitoring and sharing information with WHO. Expanded real time PCR testing capacities were introduced shortly after declaration of pandemic in all certified laboratories, hospitals, primary and community care testing facilities, accessible for all risk groups as well as symptomatic patients and close contacts. The NPIs in 2020 included obligate use of medical and personal protective equipment, restriction of mass gatherings, isolation of patients with COVID-19, PCR-testing of everyone returning from abroad, obligate temperature monitoring, social distancing, school and university closure and switch for online education and remote work for every possible position.  Lately, the NPIs were loosened or toughed again depending on COVID-19 intensity of circulation. In the described period in summer 2021, NPIs included obligate mask use in all public transport and public places, obligate vaccination of 80% of staff working at governmental organizations, QR-codes of vaccination for entry of any public place, part-closure of educational facilities and non-essential shops as well as social distancing. By October-November 2021, these were complemented by full stop of mass gatherings (more than 500 – 1500 persons), closure of food courts and  big malls. In March 2022, the mass gatherings could be resumed along with opening of shops, malls, partial freedom from use of QR-codes for entry. Since spring 2022 the NPIs were gradually loosened until summer 2022, when all COVID-19 restrictions were lifted including wearing of protective masks and social distancing.

Yours sincerely,

Anna Sominina, Daria Danilenko and co-authors

Reviewer 3 Report

Sominina et al conducted a survey based on non-sentinel influenza surveillance system in 60 major cities of Russia. They collected weekly aggregated data on incidence, hos pitalisations and deaths among patients with laboratory confirmation for influenza, SARS-CoV-2, RSV, and other respiratory viruses.

The topic is very interesting and the paper is well writen.

I suggest adding a section on limitation of this work

Author Response

Dear Reviewer 3,

Thank you very much for your hard work reviewing our manuscript and for the suggestion provided. Please see below our answer:

Reviewer 3 comment:

Sominina et al conducted a survey based on non-sentinel influenza surveillance system in 60 major cities of Russia. They collected weekly aggregated data on incidence, hospitalisations and deaths among patients with laboratory confirmation for influenza, SARS-CoV-2, RSV, and other respiratory viruses.

The topic is very interesting and the paper is well written. I suggest adding a section on limitation of this work

Response:  We fully agree that description of limitations was necessary. Please see below our insert for the corrected version of the paper.

Limitations. The analysis presented in the paper contains certain limitations, the most important of which are the limited geographic representation and use of routine surveillance data only. In particular, only sixty regions of the country were covered by the influenza surveillance system established by the NICs out of 86 in the country. In particular, only sixty regions of the country were covered by the influenza surveillance system established by the NICs out of 86 in the country. Limited by the incomplete geographical representativeness, we may have missed some circulation of influenza A(H1N1)pdm09 or influenza B, as well as minor epidemiological trends in circulation of non-influenza respiratory pathogens. Moreover, as routine surveillance has only aggregated data and does not separate severe patients from the general hospitalization statistics it was impossible to assess the contribution of influenza of different subtypes, SARS-CoV-2 and other respiratory viruses to the severity of disease in the described period.

Yours sincerely,

Anna Sominina, Daria Danilenko and co-authors
